# Empty Zona Pellucida Only Case: A Critical Review of the Literature

**DOI:** 10.3390/ijerph18179409

**Published:** 2021-09-06

**Authors:** Charalampos Siristatidis, Despoina Tzanakaki, Mara Simopoulou, Christina Vaitsopoulou, Petroula Tsioulou, Sofoklis Stavros, Michail Papapanou, Peter Drakakis, Panagiotis Bakas, Nikolaos Vlahos

**Affiliations:** 1Assisted Reproduction Unit, Second Department of Obstetrics and Gynecology, Aretaieion Hospital, Medical School, National and Kapodistrian University of Athens, 11528 Athens, Greece; dtzanakaki@gmail.com (D.T.); marasimopoulou@hotmail.com (M.S.); cvaitsopoulou@yahoo.com (C.V.); petroulatsi@yahoo.gr (P.T.); pbakas74@gmail.com (P.B.); nfvlahos@gmail.com (N.V.); 2Second Department of Obstetrics and Gynecology, Aretaieion Hospital, Medical School, National and Kapodistrian University of Athens, 11528 Athens, Greece; mixalhspap13@gmail.com; 3Molecular Biology of Reproduction Unit and Recurrent Abortions Unit, Assisted Reproduction Unit, First Department of Obstetrics and Gynecology, Alexandra Hospital, Medical School, National and Kapodistrian University of Athens, 11528 Athens, Greece; sfstavrou@yahoo.com (S.S.); pdrakakis@hotmail.com (P.D.)

**Keywords:** empty zona pellucida, oocyte retrieval, *ZP3* gene, assisted reproduction

## Abstract

The presence of empty zona pellucida (EZP) in oocytes following oocyte retrieval (OR) during an in vitro fertilization (IVF) cycle presents a major clinical and laboratory challenge in assisted reproduction. It has been attributed to several factors such as the ovarian stimulation protocol employed, the damaging of the follicles during oocyte retrieval (OR) mainly through the high aspiration pressure, during the denudation technique, and the degeneration of oolemma within the zona pellucida (ZP) through apoptosis. The role of ZP is pivotal from the early stages of follicular development up to the preimplantation embryo development and embryo hatching. Polymorphisms or alterations on the genes that encode ZP proteins may contribute to EZP. We present a critical review of the published literature hitherto on EZP and available options when encountered with the phenomenon of EZP. Concerning the former, we found that there is rare data on this phenomenon that merits documentation. The latter includes technical, genetic, and pathophysiological perspectives, along with specific treatment options. In conclusion, we identify the lack of a definitive management proposal for couples presenting with this phenomenon, we underline the need for an algorithm, and indicate the questions raised that point towards our goal for a strategy when addressing a previous finding of EZP.

## 1. Introduction

Oocyte quality constitutes a parameter of paramount importance with respect to in vitro fertilization (IVF) outcomes [1]. A conundrum that IVF clinicians are called to confront is the presence of empty zona pellucida (EZP) oocytes, referring to a finding described as the lack of ooplasm in the presence of an empty zona pellucida (ZP) [2]. This phenomenon may occur most frequently in some or, in extremely rare cases, in all oocytes during retrieval (OR). It may be attributed either to mechanical factors, namely the damage induced by applying high suction pressure during the OR, or due to the degeneration of the ooplasm itself within the ZP, possibly through the process of apoptosis [3].

The significance of the ZP throughout the process of fertilization and early embryo development is high [4]. The ZP contributes towards the fertilization process through enabling the identification of the most competent spermatozoa, the transit into cumulus cells, and the attachment on ZP receptors in order to penetrate the ZP. The latter is enabled by the acrosome reaction, which both in vitro and in vivo experiments showed that already initiates and occurs during sperm’s movement into the fallopian tube [5,6]. Subsequently, the special structure of ZP prevents the invasion of additional spermatozoa, thus avoiding polyspermy, and protects the embryos from mechanical stress prior to implantation. Thenceforth, ZP encloses the preimplantation embryo, which minimizes the risk of a potential mechanical stimulus that may provoke stress [4].

So far, studies that report on EZP oocytes are limited; these pertain to reports on IVF cycles in which a percentage of retrieved oocytes have been described as EZPs. Reports where the entire cohort of the retrieved oocytes have been EZP oocytes, a case that we herein refer to as “EZP-only”, are far fewer. The possible genetic causes implicated in EZP are largely unknown. The management of such cases is rarely described in the literature. As such, there is an urgent need for further investigation of the potential underlying molecular mechanisms entailed in ZP formation that could be involved in the clinical manifestation of EZP [5].

This study aims to present a thorough review of the respective literature, while attempting a critical evaluation of the options that should be presented to an infertile couple undergoing controlled ovarian stimulation (COH) for IVF when the entire cohort of the retrieved oocytes are classified as EZP. In this context, we underline the questions raised above and outline the possible scenarios, thus aiming to contribute to the definition of a management proposal and delineating a brief, yet based on existing evidence, proposal of management. The value of reaching a consensus on the subsequent management of such couples is also presented.

## 2. Materials and Methods

### Search Strategy

All types of studies that provide evidence on EZP in all and/or in some of the retrieved oocytes in an IVF cycle, and on potential pathways that have been proposed to explain this process, have been included. We performed the relevant literature search in April 2021 in PubMed, from 1977 to 2021, following linked references; the key words employed were the following: “oocyte quality; female infertility; zona pellucida; empty zona pellucida; mutation; ZP genes; causative gene; fertilization; in vitro fertilization; infertility; oocyte retrieval”. No restrictions pertaining to publication language or study design were adopted. Reference lists of relevant articles were hand-searched for potentially eligible studies (“snowball” procedure), on the grounds of not being a systematic review, so as to maximize the amount of the existing evidence reviewed.

## 3. Results

During the initial search, 6319 articles were detected using “zona pellucida” and searched for eligibility as aforementioned, 59 using “empty zona pellucida”, 499 using “ZP genes”, 7 either using “empty zona pellucida” AND “”genes” or “empty zona pellucida” AND “causative genes”, 20 using “empty zona pellucida” AND “infertility”, 13 using either “empty zona pellucida” AND “female infertility” or “empty zona pellucida” AND “mutation”, 266 using “zona pellucida” AND “oocyte quality”, 19 using “empty zona pellucida” AND “in vitro fertilization”, and finally, 2 using “empty zona pellucida” AND “oocyte retrieval”. The most relevant have been used as references in our review.

### 3.1. Definitions

The ZP consists of a glycoprotein stroma that encloses the oocytes and embryos up to the blastocyst stage, leading to the process of hatching, a prerequisite for the embryo’s implantation [7,8]. Considering the three-dimensional structure of ZP, the importance of ensuring the sequence of DNA bases becomes clear. Following biochemical and electron microscopic observations, the ZP appears as a heterodimeric chain of ZP2 and ZP3, which is stabilized by the homodimer ZP1 [8]. Cao and colleagues analyzed the ZP1 mutations through the structural prediction: the substitution of amino acids caused by the mutation resulted in the conversion of an arginine residue to a histidine. According to the authors, the alteration caused structural instability affecting the cross-linking or binding function of proteins. In this context, in vitro experiments in HeLa cells showed that the mutant ZP1 reduced the interaction with ZP2 and ZP3 so that the mutation influenced the cross-linking function of ZP1 and impeded the formation of ZP [8]. This indicates that possible genetic alterations could modify the structure of the ZP, leading to variations with respect to its biological function and affecting a series of processes from the oocyte development to the fertilization process [3,8].

EZP has been defined as an oocyte that either has “left” the ZP, has undergone degeneration, with the appearance of one or more breaches in the ZP and the lack of the oolemma, or is identified as a “free oocyte” near the EZP [7]. In either case, these oocytes can be visible at the time of OR or after the denudation of cumulus–oocyte complexes (COCs) prior to the intracytoplasmic sperm injection (ICSI) [2,9,10]. Studies classify EZP cases taking into account the appearance of at least one breach on ZP, which is sometimes accompanied by the identification of potential ooplasm fragments, resulting in EZP type-1 and EZP type-2, respectively [9]. The occurrence of a fractured ZP in the retrieved oocytes cohort may reflect poor oocyte quality, especially if it is observed during repetitive IVF attempts. Thus, the finding of EZP or damaged oocytes has a detrimental effect on the oocyte cohort dynamic and the subsequent fertilization potential [2,11,12,13,14], resulting in lower chances of successful fertilization, decreased numbers of embryos available for transfer, and occasional embryo transfer cancellation [11].

### 3.2. Effects of ZP Mutations and Polymorphisms

Total oocyte loss along with the presence of EZP exclusively strengthens the theory of possible genetic defects with regard to mutations in ZP genes, presenting as factors for female infertility [15]. Considerable research interest has been focused on exploring the genes of zona pellucida sperm-binding proteins 1–4, namely ZP1, ZP2, ZP3, and ZP4, identified on chromosomes 11, 16, 7, and 1, respectively. These are known to encode the four main glycoproteins, ZP1, ZP2, ZP3, and ZP4, that constitute the ZP [16]. Following sequencing analysis of the aforementioned genes, it was reported that potential polymorphisms could be associated with heterogeneous structural abnormalities of the ZP, including oocyte lysis [7]. The ZP is a glycoprotein matrix structure that encloses the oocytes and embryos until the blastocyst stage leading to the process of hatching, which is a prerequisite for the embryo’s implantation [8]. The ZP genes encoding these glycoproteins are both paralogs, possibly entailing the different function of the respective protein, and orthologs, maintaining the same function [17]. ZP is formed during the follicle development and it is vital for oocyte growth [18] as well as for the overall production of oocytes [19]. The projections of follicular cells towards the oocytes facilitates direct connections so that both contact and interactions of follicular cells with the ZP of the oocytes play a pivotal role in folliculogenesis [20].

Apart from their presence in oocytes, ZP proteins have also been detected in granulosa cells at the stage of primordial follicles, and their levels are elevated during follicular development [8,21]. It is known that the ZP proteins support communication and interaction between oocytes and follicles, protect the oocytes [22,23,24], and are produced and secreted independently [8]. Undoubtedly, the ZP plays a pivotal role in the sperm–oocyte binding process and respective interactions prior to and following the fertilization process in order to prevent polyspermy and enable the acrosome reaction [8,25]. Finally, the ZP protects the subsequent preimplantation embryo [3] and enhances the contact of embryonic cells to enable the compaction process [17].

In this context, potential alterations on genes encoding the proteins of the ZP family may exert a negative impact on the formation of the ZP, leading to either an abnormal ZP or even to its total absence [17]. Interestingly, studies demonstrated that a homozygous mutation in *ZP1* may result in EZP oocytes or even the ultimate lack of retrieved oocytes during an IVF cycle [3,19], while the homozygous variants of *ZP2* may lead to a thin ZP and hinder the fertilization procedure [3]. Mutations on the ZP genes, including a homozygous mutation in *ZP1*, heterozygous mutations in *ZP1* or *ZP3*, or a compound mutation in *ZP2,* may negatively influence the normal development of oocytes along with the normal formation of ZP, resulting in oocyte degeneration and even the empty follicle syndrome (EFS), thus compromising female fecundity [3,8,17,22,26,27,28]. As a result, ZP structural abnormalities can hinder adequate communication of COCs, impairing the oocyte quality [17].

In the literature, *ZP1*, *ZP2*, and *ZP3* gene alterations have been associated with female infertility, mainly on the grounds of ZP defects. While these women present without any ovarian or uterine anomalies with respect to menstruation (for example, they usually report normal menstrual cycles), ovulation, hormonal profile, and fallopian tube patency [29], their infertility mainly originates from oocyte maturation defects and, in some cases, with premature ovarian failure [28,30].

Mutations on the ZP genes negatively affect the normal development of oocytes along with the normal formation of ZP, resulting in oocyte lysis due to ZP defects, and in EFS, compromising female fecundity [3,8,17,22,26]. Evidence stemming from studies on ZP1-knockout mice demonstrated that the oocytes may develop a loosely organized ZP in the case of an absent ZP1 gene function [31]. Such issues pertaining to the oocyte fragility may be identified in rare cases of patients. Fragility may be identified during the ICSI procedure and it may be correlated with the phenomenon of oocyte lysis [17]. Other studies conducted on female mice reported that the absence of ZP2 and ZP3 proteins from ZP resulted in the development of ZP-free oocytes [32]. The *ZP* genes were also examined in humans, with various studies identifying that alterations in the *ZP1* gene, along with the novel mutations of ZP1 (c.1708G > A, p.Val570Met; c.1228C > T, p.Arg410Trp; c.507del, p.His170Ilefs*52), ZP1 (c.1430 + 1G > T, p.Cys478X and c.1775-8T > C, p.Asp592Glyfs*29), ZP2 (c.1115G > C, p.Cys372Ser), ZP3 (c.763C > G, p.Arg255Gly) [3], ZP1 (NM_207341:c.326G > A) [8], and ZP3 (p.Ser173Cys, c.518C > G) [25], were found to be associated with the EZP oocytes, mainly in women presenting with primary infertility. These findings highlight the need for further investigation of EZP phenomenon from a genetic perspective.

### 3.3. Defects in ZP Adversely Affects IVF Outcome

Regarding the scenario of oocyte maturation arrest, several studies demonstrated the association of oocyte degeneration with the EZP [9,33,34]. The degenerated oocytes may be associated with either an EZP or with oocyte fragments within the ZP [9,11]. Thus, an EZP finding can be indicative of oocyte apoptosis during the process of oocyte maturation. Furthermore, a retrospective study investigated the potential association of cumulus cells’ apoptosis rate with the ZP thickness and the oocyte maturity status. The results revealed a higher incidence of cumulus cells’ apoptosis in the presence of EZP oocytes or when incomplete oocyte maturation occurred [35]. Oocyte apoptosis may be also related to the EFS, which has been defined as the lack of yielded oocytes from puncturing and aspirating follicles following OR, despite the seemingly normal follicular development during ovarian stimulation, recurrent aspiration, and flushing [9,22,36]. Hence, the contributing role of oocyte apoptosis or degeneration towards the appearance of EFS has been mainly reported on the grounds of ovarian aging, adverse levels of hormones in follicular fluid (FF), genetic or metabolic changes in granulose cells, along with an impaired contact of cumulus cells with the oocyte itself [26,37]. Moreover, apoptosis of granulose cells may result in poor oocyte and embryo quality, as well as in empty follicles [38].

Available data, which is limited, documents poor IVF outcomes in cycles where abnormal oocytes were detected, namely either EZP oocytes or degenerative ones [2,11]. The study conducted by Oride and colleagues reported poor fertilization and embryo development following the comparison of cycles involving no EZP oocytes in all cycles, at least one EZP in all cycles, and those with and without EZP, revealing the pregnancy rates of 30.4%, 20%, and 28.6%, respectively [2]. Finally, IVF cycles presenting with EZP were associated with significantly lower embryo quality following the observation of morphokinetics, further highlighting the adverse effect of the occurrence of EZP on IVF outcomes [11,12,39].

### 3.4. The Role of the Ovarian Stimulation Protocol

Data in the literature has documented no significant effect of the ovarian stimulation protocols employed, including the long and the gonadotropin-releasing hormone (GnRH) antagonist protocols, on the probability of the abnormal formation of EZP oocytes [13]. This general observation is in discordance with the findings of the study conducted by Cinar and his colleagues, which reported a higher proportion of the number of EZP to the number of COCs retrieved when employing the GnRH antagonist protocol; the latter was associated with a negative effect of the IVF protocol, causing oocyte degeneration. As the reporting of EZP is rare in the literature, we are currently uncertain if the protocol (mild, use of GnRH agonist, or antagonist), the dose of any regimen (including gonadotrophins), the duration, the timing of oocyte triggering and oocyte recovery and/or oocyte preparation, and, finally, the manipulation can cause the phenomenon. What is more, sufficient ovarian stimulation with increasing estradiol levels has been positively associated with good oocyte quality and negatively associated with EZP [9,40], indicating that estradiol is an important factor for oocyte maturation and quality [9,41]. Thus, inadequate stimulation of granulosa cells may result in EZP formation [9,40]. Animal studies further support these theories: Visser and colleagues, using null mice to knock-out the anti-Mullerian hormone (AMH) gene, demonstrated that AMH levels could be related to the oocyte degeneration or the EZP formation [9,42]. They further exhibited that there are many oocyte remnants in the ovaries of these mice, reflecting the occurrence of oocyte apoptosis at the early stages of follicular development.

### 3.5. Management Approaches

Following the identification of the EZPs-only finding, the IVF team should thoroughly discuss the possibilities and scenarios of how the EZP oocytes resulted, concurring on several technical parameters.

In Figure 1, we present a brief summary of the recommended steps. These include the reassessment and potential change of: the technical procedure around the oocyte retrieval (from both clinicians and embryologists); the hormonal characteristics of both partners, including the ovarian reserve of the woman; and the COH protocol implemented during the cycle where EZP was observed, including any regimens applied, time, dose, and adjuncts used. All the aforementioned parameters have been meticulously reported in the literature as being potential contributing factors towards the occurrence of EZP oocytes. However, this critical review points out the lack of data agreeing upon effective management of “EZP-only” cases, which further highlights the imperative need of conducting well-designed studies. In addition, a proper consultation by the adequate personnel should include genetic analysis of the ZP1-4 genes linked to bioinformatics correlations prior to considering a second IVF attempt, along with couples’ further options, such as oocyte donation and adoption.

### 3.6. Causative Factors and the Relevant Background

The plausible initial explanations for the phenomenon of EZP may entail the following: the small diameter of the employed needle during the OR [10,31,36], the high aspiration pressure used, and the size of the denudation pipettes “stripper tips” employed during the OR and denudation procedures [43,44]. Regarding the latter, if the stripper tips employed for the denudation process are close to the diameter of an oocyte (120 μm), damaging the ZP or stressing the oocyte would be a possibility that should be considered. Interestingly, the link between oocyte aspiration procedures and oocyte damage was reported long ago: according to the experiments of Bols and colleagues (1997), the length of the needle bevel exerted a significant effect on oocyte recovery, in favor of the long-beveled needle [10]. As soon as higher-aspiration vacua were used, a decrease in the number of oocytes was observed, which was less prominent for the short-beveled needle compared to the long-beveled one. In addition, at low aspiration vacuum, up to 90% of the oocytes remained undamaged, while when increasing the aspiration vacuum, there was a decrease in the number of oocytes, which was less pronounced using thinner needles [10].

### 3.7. The Main Recommendation: Genetic Analysis and Bioinformatics

Couples should be advised to proceed to genetic analysis of the genes ZP1-4, to examine potential mutations that may lead to EZP oocytes, prior to considering a second IVF attempt. Usual techniques involved are whole-exome sequencing (WES) for genes that were directly sequenced (ZP1, ZP2, ZP3) in combination with bioinformatics analysis (using specific databases) to analyze and compare the sequences detected.

Quoting the recent paper by Normand and colleagues (2018), WES is a powerful and unbiased tool for identifying genetic variation by capturing genome-coding regions. It covers a region of approximately 1–1.5% of the human genome, where approximately 85% of causative mutations are located. Overlap-, de novo-, rare phenotypes, and familial-based are the four main strategies for WES analysis. This is advantageous for diagnosing patients in whom a genetic disorder is suspected but who have a nonspecific or atypical phenotype, or when the disorder has significant genetic heterogeneity [45].

The current target in preconception medicine for utilizing WES consists of patients with recurrent pregnancy loss or unexplained infertility. The main advantage is the prevention of the so-called “diagnostic odyssey”, protecting both couples and clinicians from the loss of time, money, stress, and uncertainty. Using one test to assay all known genes simultaneously has the potential to return a molecular diagnosis in a quicker and more cost-effective manner than performing an extensive clinical workup of the “unknown”. Examples of medical management changes include therapy modification, cessation, or initiation and surveillance changes for disease-associated complications. Additional clinical benefits include the modification of recurrence risk estimates for future pregnancies and the ability to use preimplantation genetic diagnosis, donor gametes, or targeted genetic testing for future pregnancies. In addition, even in cases where medical management is not affected per se, as the information itself may not immediately result in clinical utility, they can eventually prove useful for patients and clinicians towards the discovery of novel gene-related diseases [45].

Additionally, a further help from bioinformatics would be essential, such as omics technology. Quoting the paper of Altmäe et al. (2014), “Omics” refers to the application of high-throughput techniques that simultaneously take into consideration the alterations in the genome, epigenome, transcriptome, proteome or metabolome in a certain biological sample [46]. There are also new fields in biological data, such as exomics, lipidomics, and secretomics. “Omics” data are a valuable parameter for embryo selection optimization and promoting personalized IVF treatment [47]. For example, proteomics is an emerging, powerful diagnostic tool, with applications ranging from biomarker identification for effective embryo and oocyte selection and high-risk pregnancy management, to tissue engineering guidance [48]. Another example is transcriptomics; the concept behind this is that gene expression in CCs engulfing the oocyte provides crucial information on the competence of the oocyte itself and, furthermore, the embryo, as these two differential cell-types grow and develop in a highly coordinated and mutually dependent manner. Gene expression analysis of CCs is achieved through the isolation of these cells before ICSI or IVF and the analysis of the extracted mRNA through microarray analysis, RT-PCR, or quantitative RT-PCR to obtain its transcriptomic profile [47].

Finally, another potential source of bioinformatics derives from the knowledge of the levels of Reactive Oxygen Species (ROS) in the FF after the OR. ROS are produced as a normal product of aerobic metabolites. ROS exert direct and indirect effects on the dynamics of the oocyte in terms of maturation and fertilization potential, either spontaneously or through assisted reproduction technologies. The adverse relationship between the ROS levels in the FF and oocyte maturation and fertilization is seen as a logical explanation of an oocyte with low dynamics. In an IVF setting, the existing literature suggests a favorable outcome in terms of oocyte quality/maturation and fertilization rate with increased ROS levels, while other studies report significant data on the detrimental effect of increased ROS concentration in the quality of embryos exposed and their potential to advance [49,50]. It appears that an optimum level of ROS is necessary to support adequate oocyte development, whilst the notable imbalance in the ovarian environment following controlled ovarian hyperstimulation results in excessively high ROS levels and, consequently, in oxidative stress [51]. Data reveal the exceptional role of favorable ROS levels on folliculogenesis process, highlighting the association of oxygen concentration with the proper follicular angiogenesis. Follicular development from the primordial to antral stage consists of a highly metabolic process, producing ROS. The aforementioned indicates the importance in maintaining favorable levels of ROS in an FF environment [38,52,53].

Furthermore, in a recent paper from the authors’ team of the current study, “omics” combined with predictive models have been suggested to have the ability to substantially promote health management individualization and contribute to the successful treatment of infertile couples, particularly those with unexplained infertility or repeated implantation failures and, in rare cases, those currently reported in this review [47].

### 3.8. Further Recommendations

Apart from the technical, genetic, and pathophysiological perspectives mentioned above, the main issue pertains to the treatment options available following a finding of EZP-only oocytes. These include proceeding to a subsequent cycle considering previously suggested actions and the recommendation of an oocyte donation program. The latter is usually opted for in cases presenting with defects in the oocyte maturation process, whereas adoption could represent the last resort. Of note, it appears that literature has not a lot to offer on presenting practitioners with a universally accepted approach concerning the efficient management for these cases, a fact adding another level of complexity to their subsequent management.

Future research should focus on how a subsequent cycle’s effectiveness for couples featuring EZP or EZP-only oocytes could benefit from modifications in the protocols employed. These should focus on the investigation of alternative IVF protocols, such as natural IVF cycles, mild stimulation protocols with the administration of Clomiphene Citrate or aromatase inhibitors with or without gonadotrophins, and the use of oral contraceptives for a time period of three months before the next IVF treatment and possible differentiation and adjustments afterwards. These could all be options in subsequently treating these couples and achieving better results. Moreover, and with regard to the OR and the role it may play, practitioners may consider employing the use of a wider needle and lower vacuum pressure, for instance, 80–100 mmHg in a subsequent cycle. Finally, the need to delve into identifying the molecular backbone of the ZP glycoprotein structure and performance is highlighted herein. The need for an informed consent form detailing information about subsequent treatment for these couples and examining what one can expect from a second IVF cycle should be examined. Finally, and most importantly, it is of paramount importance to highlight that the lack of a universally established protocol in addressing rare events, such as IVF cycles involving EZP and EZP-only, may harbor the risk of empirical mismanagement.

## 4. Conclusions

The exclusive finding of EZPs for the entire cohort of retrieved oocytes has not been reported hitherto in the current literature. Even for cases featuring some EZPs, little is known about the underlying causes, or the subsequent efficient management, of this distinct group of infertile couples [2,7]. Although the underlying etiopathogenesis remains unclear, genetic factors seem to play a prominent role, while technical parameters or the choice of the ovarian stimulation protocol might be involved. Affected couples should be therefore advised to undergo genetic analysis, specifically targeting the ZP1–4 genes [15,16].

Inevitably, the main limitation in what we know derives from the fact that there is low-quality evidence stemming from the nature of the types of studies reporting on this rare phenomenon [3,26]. Undoubtedly, completion of larger studies encompassing the genetic perspective will be of merit. The scientific community should further focus on providing well-defined guidelines in an effort to describe the optimal management of couples presenting with EZP, along with defining the strategy for handling subsequent IVF cycles. Nonetheless, guidelines could only be provided following robust data and, in light of the fact that EZP is a rare phenomenon, it may be challenging for such data to be sourced [3]. Currently, management of patients presenting with an EZP finding appears to be empirical. However, in the era of precision medicine, every effort should be made to avoid empirical mismanagement. When confronted with an EZP finding, the assisted reproduction specialist is uncertain of the optimal management in addressing this conundrum. Should we counsel the couple to undergo genetic analysis? Should they undergo a subsequent IVF cycle irrespective of genetic analysis feedback, and if so, how many attempts are advised and on what grounds prior to suggesting oocyte donation? Is there a place for a dedicated consent form when proceeding with a subsequent IVF treatment? Answering these questions raised will in turn enable formation of an algorithm for proper management. In Figure 1, authors outline all aspects that should be considered when encountering an EZP finding in an effort to facilitate and provide guidance for both clinicians and researchers. Reassessing COH [13] and OR [3] procedure, along with laboratory handling, may provide an explanation for the incidence of EZP and serve as an indicator regarding what to avoid and which COH and OR procedure to adopt in a future cycle. Furthermore, genetic analysis targeting ZP1–4 genes should be considered as studies have underlined the genetic aspect implicated in this finding [16]. Nonetheless, despite the benefit of genetic analysis, how to further counsel the patient is not yet defined. What becomes clear is the value of counselling the patients prior to discussing the option of oocyte donation or adoption in the aim of enabling informed consent and providing patients with high-quality services in assisted reproduction treatment.

## Figures and Tables

**Figure 1 ijerph-18-09409-f001:**
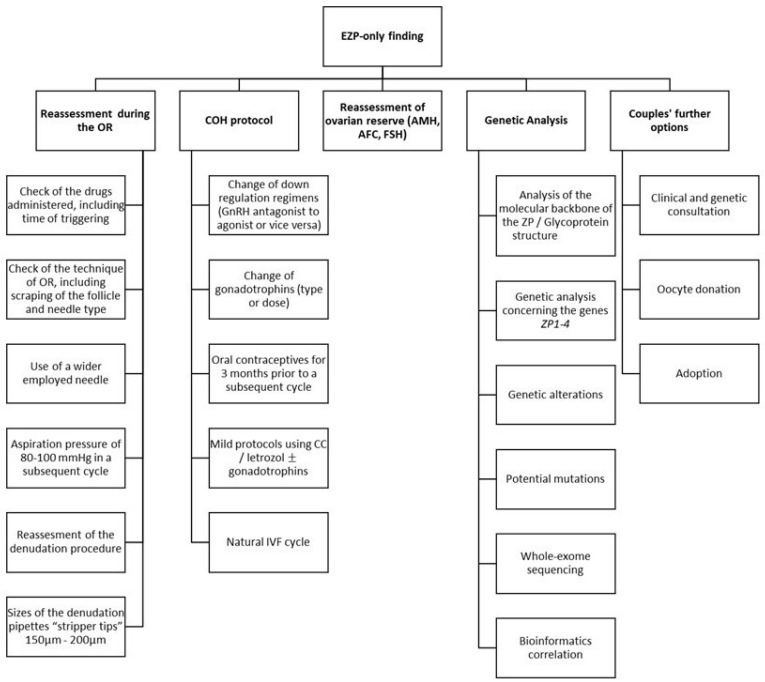
Summary of recommendations for the management of patients presenting with the empty zona pellucida (EZP)—only finding. Abbreviations: EZP, empty zona pellucida; OR, oocyte retrieval; COH, controlled ovarian hyperstimulation; GnRH, gonadotropin-releasing hormone; CC, clomiphene citrate; IVF, in vitro fertilization; AMH, anti-Mullerian hormone; AFC, antral follicle count; FSH, follicle stimulating hormone; ZP, zona pellucida.

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
