# Peer review of "Empty Zona Pellucida Only Case: A Critical Review of the Literature"

_ijerph, 2021, doi:10.3390/ijerph18179409_

Round 1

Reviewer 1 Report

The review is on an interesting topic, but this reviewer suggests the following changes:

  1. The manuscript needs editing for grammar. For example, the text is missing the word "the" in several places.  Some sentences need to be reworded for clarity.
  2.  In the methods, the authors should indicate how many manuscripts were identified in their search strategy. The topic is very narrow so this reviewer is wondering about the actual number of manuscripts identified in the search. 
  3. The authors present a number of recommended steps in figure 1. Are these steps based on the literature or the authors experience? The authors should clearly indicate why they are recommending certain steps.
  4.  The authors should "back-up" some of their suggestions in the conclusion section with references. It is not clear if the authors are offering their own opinion and if this is an opinion article or if they are making recommendations based on the literature. 

Author Response

The review is on an interesting topic, but this reviewer suggests the following changes:

A: The authors are grateful for the time and effort in reviewing this manuscript and the constructive comments aiming to improve the overall performance of this manuscript. Please kindly note that revisions are highlighted using the “Track Changes” function throughout the text. Moreover, you will find the reference list altered. Herein follows the point-by-point response.

1. The manuscript needs editing for grammar. For example, the text is missing the word "the" in several places.  Some sentences need to be reworded for clarity.

A: We totally agree; we have done this throughout the whole text. Please see the track changes version.

2. In the methods, the authors should indicate how many manuscripts were identified in their search strategy. The topic is very narrow so this reviewer is wondering about the actual number of manuscripts identified in the search. 

A: Thank you for highlighting this point of high interest. The number of manuscripts detected following the combination of the keywords used were varied from 6319 when searching for zona pellucida and 59 for empty zona pellucida to 7 when searching for empty zona pellucida AND genes. This difference further points out the few evidence existing in the literature as you mentioned. Please see revised version, page 3.

3. The authors present a number of recommended steps in figure 1. Are these steps based on the literature or the authors experience? The authors should clearly indicate why they are recommending certain steps.

A: Thank you for this aptly raised comment. The authors aimed to critically evaluate the existing literature regarding the options for managing empty zona pellucida cases in the context of an IVF cycle. Thus, Figure 1 proposes certain steps based on data existing in the literature that report the contribution of several parameters-for example mechanical, genetic- towards the occurrence of empty zona pellucida oocytes. This point has now been clarified in the text. Please see revised version, page 6 and 10.

4. The authors should "back-up" some of their suggestions in the conclusion section with references. It is not clear if the authors are offering their own opinion and if this is an opinion article or if they are making recommendations based on the literature. 

A: Thank you for pointing this out. References have now been provided. Please see revised version, page 10.

Reviewer 2 Report

This is an interesting review article that deals with an important topic (EFS) that has not been much covered by previous reviews. Therefore, the timeliness of the Manuscript seems to be appropriate. In addition, the Manuscript is easy to follow and I just have some specific comments.

SPECIFIC COMMENTS

The Manuscript would benefit from adding a diagram with the oocyte, indicating the potential links between EFS and its causes.

Introduction

These sentences “The significance of the ZP throughout the process of fertilization and early embryo development is high [4]. Indeed, ZP contributes to-wards the fertilization process through enabling identification of the most competent spermatozoa, attachment on ZP receptors and activation via the acrosome reaction, which enables ZP penetration” should be revised. In effect, previous research by Buffone and Visconti suggest that acrosome reaction is not triggered upon interaction between sperm and ZP, but before in the fallopian tube. Therefore, this section should be revised, at least making a reference to the in vivo and in vitro scenarios.

“Management of such cases is rarely described in the literature, further highlighting the need for further investigation of the potential underlying molecular mechanisms entailed in ZP formation that could be involved in clinical manifestation of EZP [5].”: avoid the use of ‘further’ twice in the same sentence.

M&M

Please indicate when the search was conducted and which years covered (i.e. papers published from XXXX to 2021)?

It would be nice of the authors could provide a table/diagram with the search strategy

Results

The relationship between Oocyte apoptosis and EFS would benefit from being extended a bit, as it seems to be very important.

Bols and colleagues: please cite the paper in compliance with the Journal guidelines

I also suggest to extend a bit the paragraph intended to ROS and EFS

Author Response

The authors appreciate your constructive review and direction you have provided in highlighting points of interest. Please kindly note that revisions are highlighted using the “Track Changes” function throughout the text. Moreover, you will find the reference list altered. Herein follows the point-by-point response.

SPECIFIC COMMENTS

The Manuscript would benefit from adding a diagram with the oocyte, indicating the potential links between EFS and its causes.

 A: Thank you for this suggestion. The subject of interest was specifically focused on EZP; for its causes, we have made specific comments; we did not proceed to analyze the causes of EFS. If the reviewer insists on providing such a table, we would be more than happy to proceed to that, in a revised version.

Introduction

These sentences “The significance of the ZP throughout the process of fertilization and early embryo development is high [4]. Indeed, ZP contributes to-wards the fertilization process through enabling identification of the most competent spermatozoa, attachment on ZP receptors and activation via the acrosome reaction, which enables ZP penetration” should be revised. In effect, previous research by Buffone and Visconti suggest that acrosome reaction is not triggered upon interaction between sperm and ZP, but before in the fallopian tube. Therefore, this section should be revised, at least making a reference to the in vivo and in vitro scenarios.

A: Thank you for highlighting this point of high importance. The respective sentences have now been amended. Please see revised version, page 2.

“Management of such cases is rarely described in the literature, further highlighting the need for further investigation of the potential underlying molecular mechanisms entailed in ZP formation that could be involved in clinical manifestation of EZP [5].”: avoid the use of ‘further’ twice in the same sentence.

 A: Thank you for highlighting this. It has now been amended. Please see revised version, page 2.

M&M

Please indicate when the search was conducted and which years covered (i.e. papers published from XXXX to 2021)?

It would be nice of the authors could provide a table/diagram with the search strategy

 A: Thank you for indicating this. The search was performed in April 2021 without a date restriction, namely all articles published in PubMed from 1977 to 2021 were identified and manually screened. Moreover, on the grounds of not being a systematic review but a critical one, a paragraph including the number of articles detected following several searches in PubMed is now added in materials and methods instead of a diagram/table. The authors trust you will find the respective changes complete. Respective data is now provided. Please see revised version, pages 2-3.

Results

The relationship between Oocyte apoptosis and EFS would benefit from being extended a bit, as it seems to be very important.

A: Thank you for highlighting this. It has now been amended. Please see revised version, page 2.

Bols and colleagues: please cite the paper in compliance with the Journal guidelines

A: Thank you for indicating this. It has now been addressed. Please see revised version page 7.

I also suggest to extend a bit the paragraph intended to ROS and EFS

A: Thank you for your suggestion. It has now been amended. Please see revised version, page 9.

Round 2

Reviewer 1 Report

The authors responded to my previous concerns.